# Glycomimetic Peptides as Therapeutic Tools

**DOI:** 10.3390/pharmaceutics15020688

**Published:** 2023-02-17

**Authors:** J. Kenneth Hoober, Laura L. Eggink

**Affiliations:** Susavion Biosciences, Inc., 1615 W. University Drive, Suite 132, Tempe, AZ 85281, USA

**Keywords:** CLEC10A, dose response, glycomimetic peptides, eczema, transglutaminase 2

## Abstract

The entry of peptides into glycobiology has led to the development of a unique class of therapeutic tools. Although numerous and well-known peptides are active as endocrine regulatory factors that bind to specific receptors, and peptides have been used extensively as epitopes for vaccine production, the use of peptides that mimic sugars as ligands of lectin-type receptors has opened a unique approach to modulate activity of immune cells. Ground-breaking work that initiated the use of peptides as tools for therapy identified sugar mimetics by screening phage display libraries. The peptides that have been discovered show significant potential as high-avidity, therapeutic tools when synthesized as multivalent structures. Advantages of peptides over sugars as drugs for immune modulation will be illustrated in this review.

## 1. Introduction

The extracellular domain of most proteins in the cell membrane is extensively decorated with carbohydrate groups. The plethora of glycans is matched by the vast diversity of lectin-type receptors. Single glycans bind with low affinity to these receptors but occur at a density sufficient to maintain physiological homeostasis. Nevertheless, to modulate an immune response under this canopy of sugars requires high-affinity ligands to activate or inhibit cellular processes. Modulators of the immune system with high activity often have a sense of multivalency, if not actually structurally multivalent. The chemistry of synthesis of multivalent glycan structures has advanced rapidly over the past few years [1,2,3,4,5,6,7]. However, glycans generally have poor drug-like properties. These concerns led to extensive chemical modifications of sugars and sugar-like structures that can serve as “glycomimetic” drugs [8,9]. Yet, it is useful to synthesize peptide mimetics of the native glycan ligand, which provide a relatively high specificity along with a high avidity of binding to receptors [10,11]. The sequences of glycomimetic peptides have usually been discovered by screening phage display libraries with lectins, a highly effective method of identifying peptides that bind to sugar-binding sites [12,13,14,15,16,17,18]. 

Protocols for screening libraries were described by Matsubara [12] and Yu et al. [13], who remarked that “these sequences would never have been predicted rationally nor would molecules with similar modes of binding have [otherwise] been discovered.” Structures larger than tetravalent are often antigenic, and an active area of research has been the search for peptide vaccines that induce the immune system to produce antibodies against viral or tumor-associated carbohydrate structures [19]. Whereas a vaccine initiates the pathway in T cells but is not the target to which the final antibodies bind, in other uses the multivalent glycomimetic peptides are direct ligands for receptors. Numerous articles and reviews of the roles of peptides as endocrine mediators and drug candidates that target various physiological pathways have been published [20,21,22]. This article will emphasize research with peptides that mimic sugar ligands of lectin-type receptors expressed by cells of the immune system and have been studied in model systems as tools for potential therapeutic use; but first we review a fascinating naturally occurring peptide that interacts with a sugar-binding site with very high avidity.

## 2. Tendamistat—An Inhibitor of α-Amylase

Among the earliest known non-glycan structures that bind to a glycan binding site is a microbial inhibitor of α-amylase, an enzyme that hydrolyzes linear polymers of glucose such as glycogen and starch to the disaccharide maltose. A small protein of 74 amino acids, tendamistat, is a tight binding inhibitor (K_i_ = 9 × 10^−12^ M) [23]. The sequence Tyr^15^-Gln-Ser-Trp-Arg-Tyr^20^ within the protein forms a loop (Figure 1A) that binds in the catalytic site of porcine α-amylase [24] (Figure 1B). We tested with CABS-Dock software [25,26] whether in silico modeling would predict the binding revealed by crystallography [24]. CABS-Dock software searches without prior assignment for a binding site of a peptide on the crystal structure of a protein and defines the most probable secondary structure of the peptide. The docking program ranks the 10 most likely models based on cluster density and average root mean square deviation (RMSD), a measure of the fit, after 50 simulation cycles. The models were downloaded into ArgusLab software, with which binding energies can be predicted. Indeed, although the 6-mer sequence is expected to be flexible in solution, the CABS-Dock software inserted it into the binding site on α-amylase as a loop, which recapitulated the crystallographic data (Figure 1C). The conformation in the crystal structure of the complex with α-amylase suggested an induced-fit type of binding of the loop structure [27]. The relatively high predicted binding energy suggested that binding of the loop structure was primarily responsible for the very high avidity of tendamistat. The conformational constraint on the loop by the remainder of the protein structure reduces the unfavorable entropic contribution and therefore increases the avidity of binding [28,29,30]. 

This type of binding is a common feature of the interaction of peptides with sugar-binding sites. Sugar residues generally occur in one of two possible conformations, boat or chair, which bind to and stabilize pre-existing receptor conformations, as proposed by Changeux [31]. The high flexibility of peptides allows binding to follow a complementary induced-fit mechanism by which conformational selection by both receptor and ligand leads to the most stable structure. However, binding of a flexible peptide imposes an entropy penalty to affinity, which is offset by the rigidity of the tendamistat structure. In the following discussion, incorporating the peptide into a multivalent structure reduces the unfavorable entropy contribution by tethering one end to the structural core and also increases avidity as the result of the ligand cluster effect (increased local concentration) [32,33,34,35]. A 10- to 20-fold increase in avidity to several lectins was demonstrated with a series of mono-, bi- and tetravalent peptide structures [36,37]. In this article, in silico modeling assumes binding by a flexible peptide ligand to a rigid crystal conformation of the receptor.

## 3. Major Families of Receptors on Immune Cells and Their Ligands

### 3.1. A Peptide Mimetic of Mannose

Complex mannose (Man)-rich glycans are linked to the amide N of asparagine in proteins. Man-containing structures are ligands for receptors such as the Man receptor (MR, CD206)) and dendritic cell-specific intercellular adhesion molecule-3-grabbing non-integrin (DC-SIGN, CD209) [38]. The earliest studies to find a mimetic of Man involved screening phage display libraries with concanavalin A (ConA), a well-studied lectin with a preference for binding Man. A series of peptides was identified with a consensus motif YPY [39,40]. A dodecapeptide, DVFYPYPYASGS, bound to ConA with a K_D_ of 46 μM, a value that indicated a slightly higher affinity than the K_D_ of 89 μM for α-methyl-Man. Binding of the 12-mer peptide to ConA was inhibited by α-methyl-Man, and the peptide inhibited the precipitation of ConA by dextran [40], which suggested that the peptide and sugar competed for the same binding site [41]. Interestingly, antibodies raised against either the 12-mer peptide conjugated to diphtheria toxin or α-methyl-Man conjugated to bovine serum albumen cross-reacted with the other antigen [42], which suggested functional similarity of the mimetic peptide and Man. However, shorter octapeptides and the tripeptide YPY bound to ConA but did not displace a chromogenic Man derivative nor did these peptides inhibit the dextran precipitation reaction [40]. Scott et al. [40] identified a hexapeptide, MYWYPY, that bound to ConA with a K_D_ of 800 μM and inhibited the precipitation of dextran by ConA with an IC_50_ of 490 μM. Crystallographic analysis showed that the peptide bound at two sites adjacent to, but not in, the Man-binding site [43], which suggested that the degree of competition between the sugar and the peptides was based on overlap from the adjacent binding sites or through conformational changes and not physically occupying the sugar-binding site. Analysis of the conformation of different peptides bound to ConA [44] showed that they achieved similar structures, an example of induced fit as proposed by Changeux [31]. In silico studies with the docking program CABS-Dock suggested that the peptide MYWYPY did not bind tightly to either site on ConA, as indicated by the RMSD values, but the predicted binding energies were the same for both sites (Figure 2). The binding site for Man was near a loop in the protein structure between the two peptide binding sites. Adding an acetyl group to the N-terminal amino group and an amide at the C-terminus (Ac-MYWYPY-NH_2_) eliminated the charge on the peptide and improved the complementarity of the surfaces. The reduction of electrostatic hindrance dramatically increased affinity to the nonpolar surface of the lectin and allowed the modified peptide to directly overlap the binding site for α-methyl-Man [45]. 

Because Man-containing glycans are attached to envelope proteins of HIV, Pashov et al. [18] examined mimetic peptides of Man for vaccines effective against the virus. Octavalent peptides based on the sequence YRYPY bound to ConA with slightly higher avidity, with a K_D_ of approximately 1 nM, than YPYPY. Human IgG that recognized RYRY and YPYRY also bound to HIV gp120. This interaction was inhibited by α-methyl-Man but also by fucose and GalNAc. However, an antibody against gp120, 2G12, bound to the PYPY sequence but not to RYRY. Thus, peptides with high affinity in lectin binding studies do not always express full mimetic properties. Kieber-Emmons et al. [19] described the potential for clinical use of mimetic-based antigens, but although many lessons have been learned in a search for vaccines, much work remains to be carried out.

### 3.2. Peptide Mimetics of Sialic Acid

An abundant protein of the red blood cell membrane is glycophorin A (CD235a), which has 17 sialic acid (5-acetylneuraminic acid, NeuAc)-containing glycans attached to the extracellular domain. The major O-linked glycans are the tetrasaccharide NeuAcα(2-3)-Galβ(1-3)-[NeuAcα(2-6)]-GalNAcα1-*O*Ser/Thr (about 78%) and the trisaccharide NeuAcα(2-3)-Galβ(1-3)-GalNAcα1-*O*Ser/Thr (17%). The structure of the tetrasaccharide is illustrated as



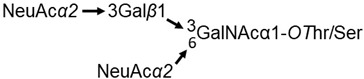



The purpose of glycophorin A is to provide a coating of negatively charged sialic acid to the cell surface, most likely to prevent clumping of cells. There are approximately 3 × 10^5^ copies of this protein per erythrocyte [46]. Most serum glycoproteins also bear sialic acid-containing glycans. Over time, the sialic acid residues are gradually lost from circulating cells and serum glycoproteins, thus exposing the galactose (Gal) or, further, the *N*-acetylgalactosamine (GalNAc) residue. The resulting glycan structures lacking sialic acid are ligands for the phagocytic receptor asialoglycoprotein receptor 1 (ASGR1) that initiates phagocytosis and degradation by hepatocytes [38,47,48].

The glycan structure on glycophorin A illustrates the primary ligands for the three major families of sugar-binding, lectin-type, receptors: siglecs, galectins, and C-type (Ca^2+^-dependent) CLECs. Siglecs (sialic acid-binding immunoglobulin-like lectins) are relatively specific for sialic acid. Of the structurally related family of 14 siglecs in humans, 13 are expressed on cells of the immune system [49,50,51]. A family of 15 lectin-type receptors, designated galectins, binds β-Gal residues on proteins. These receptors are involved with the modulation of many physiological functions including inflammation, immune responses, cell migration, autophagy, and cellular signaling [52,53,54]. C-type lectin receptors form a large, diverse family of 86 structurally related proteins in humans, arranged into 16 groups, that bind many different terminal sugars [55,56]. Most C-type lectin-like receptors are endocytic, transmembrane, pattern recognition receptors, expressed primarily by myeloid cells and require Ca^2+^ to bind a sugar ligand. 

The disaccharide Galβ(1-3)GalNAcα1-*O*Ser/Thr is referred to as the Thomsen-Friedenreich (T or TF) antigen. Further removal of Gal from the glycan yields GalNAcα1-*O*Ser/Thr, the Tn antigen, which is a highly specific ligand for CLEC10A (Ca^2+^-dependent lectin-type receptor family member 10A, CD301), which is expressed by dendritic cells (DCs) and macrophages. The Tn antigen is rare on cells of healthy tissues but is a marker of many tumor cells. A total of 96 [57] or 97 [58] glycoproteins bearing one or more Tn antigens was identified on human T lymphoblastoid cells (Jurkat cell line). An exception is the protein tyrosine phosphatase CD45, which is an essential protein on cells of the immune system. CD45 has two GalNAc-*O*Ser structures in the B exon of the protein. Splicing of the mRNA for CD45 during activation of T cells yields shorter isoforms that lack exon B [59,60,61]. These antigens bind CLEC10A with low avidity (K_D_ values of μM to mM) and also tend to produce antibodies with low affinities. Although a desirable target for the development of anticancer therapies, the low affinity of the Tn antigen hinders their usefulness. 

Matsubara [12,22] provided an excellent and detailed description of the protocols involved in the discovery of functional peptide mimetics of sialic acid by panning phage display libraries. He described an overview of the properties of randomized libraries and provided a list of peptides currently identified as sugar mimetics that bind as ligands for lectins or enzymes and those used as vaccines for the induction of anti-glycan antibodies. He identified a series of 15-mer peptide mimetics of sialic acid-Gal that were synthesized as stearyl conjugates, which resulted in multivalency by self-assembly of the alkyl groups in aqueous solutions. These alkyl-peptide assemblies inhibited infection of MDCK cells by influenza virus by binding to viral hemagglutinin (HA) with IC_50_ values of 8 to 11 μM. The initial 15-mer peptide was refined to a pentapeptide Ala-Arg-Leu-Pro-Arg (ARLPR), which had a dramatic increase in avidity of binding when synthesized as a tetravalent structure from a tri-lysine core, a design developed by Posnett et al. [62]. The avidity of the peptide increased from K_D_ = 74 μM as a monomer to 0.4 μM as the tetravalent structure. Conformational flexibility is constrained when one end of a peptide is tethered in a multivalent structure, which can contribute significantly to the entropy of binding. 

Of the 13 siglecs expressed by the immune system, 9 are inhibitory receptors and include an immunoreceptor tyrosine-based inhibitory motif (ITIM) and an immunoreceptor tyrosine switch motif (ITSM) within the sequence of the cytoplasmic domain. In this sense, siglecs act as checkpoint inhibitors [63]. To overcome suppression of cellular activity, a peptide was designed that bound with high specificity to sialic acid-binding lectins and with high avidity to most siglecs [36,37,63]. To find this sequence, a large number of peptide sequences was obtained from screens with GalNAc-specific lectins of a constrained (C-X_7_-C) phage display library. Because no consensus emerged, the recovered sequences were analyzed with an algorithm to search for frequent pairs. Building from this analysis, the most common sequence was a 5-mer HPSLK. When synthesized as a tetravalent peptide from a tri-lysine core, the 5-mer bound equally well to several lectins of different specificities. This peptide, designated sv6B, was a potent stimulator of phagocytosis by macrophages [36,37]. Further modifications at the N- and C-termini of the peptide, with binding activity analyzed by in silico modeling and directly with sugar-specific lectins, led to the sequence NPSHPSLG. The tetravalent peptide, designated svH1D, bound slightly to lectins that are specific for Gal, GalNAc, or GlcNAc but strongly to sialic acid-specific lectins. However, binding was detected only with lectins specific for sialic acid when the SL pair in svH1D was inverted to NPSHPLSG, designated svH1C [63]. 

Incubation of PBMC cultures with svH1C caused a rapid (within 5 min) dephosphorylation of siglecs, an indication of a reduction of the inhibitory activity of the receptors [63]. At a concentration of 50 nM, svH1C dramatically increased antibody-mediated phagocytic activity by adherent cells, a characteristic of macrophages. Internalization of microspheres that were coated with HIV-1 envelope gp41 and opsonized with anti-gp41 or coated with streptavidin and opsonized with anti-streptavidin, was dramatically increased after incubation with svH1C [36,37]. These results suggested that svH1C would be suitable as an anti-viral drug, which was confirmed by the essentially complete inhibition of replication of HIV-1 in PBMC cultures at concentrations of the peptide in the range of 1 to 10 nM [36]. These data led to the conclusion that enhancement of immune cell activity by svH1C results from lessening of the inhibitory activity of siglecs that are expressed by macrophages. 

In silico modeling with CABS-Dock showed that the predicted binding of an arm of tetravalent svH1C occurred at the sialic acid-binding site of human recombinant siglec receptors. Because the sialic acid mimetic sequences described by Matsubara et al. [22] (ARLPR) and Eggink et al. [63] (NPSHPLSG) have minimal homology, the sequences were compared by in silico binding to the crystal structures of Siglec-5 [64] and Siglec-7 [65]. Both peptides bound to Siglec-7 in the same site and with similar predicted binding energies (Figure 3). However, the two peptides bound at different sites on Siglec-5. NPSHPLSG bound in the binding site for sialic acid as determined by crystallography, whereas with the most highly ranked model for ARLPR, the peptide was bound in the “hinge” region between the two domains of the receptor with a relatively low predicted binding energy (RMSD = 3.295 Å; ΔG′ = −26.7 kJ/mol). The model that was ranked sixth had the peptide ARLPR within the sialic acid-binding site but with an unfavorable RMSD = 14.47 Å. As shown in Figure 4, these binding sites overlapped but were not identical. Although the initial binding site identified by the software for ARLPR was not the most energetically favorable, the eventual or perhaps final site based on binding energies may likely be within the binding site for sialic acid. It is interesting that these two very different sequences show similar mimetic properties.

### 3.3. Peptide Mimetics of N-Acetylgalactosamine 

A dodecapeptide, VQATQSNQHTPR, was discovered through a screen of a phage display library with the GalNAc-specific lectin from *Helix pomatia* [66]. The peptide designated svL4, when synthesized as a tetravalent structure, bound GalNAc-specific recombinant human receptors, ASGR1 and CLEC10A with K_D_ values of approximately 0.1 μM [67]. Because the peptide extends beyond the binding site, the two halves of the 12-mer were synthesized separately. The C-terminal hexapeptide, NQHTPR, contained the GalNAc-binding activity and as the tetravalent peptide, designated sv6D, inhibited binding of GalNAc-PAA to these receptors with an IC_50_ of 80 nM. However, in cell cultures, maximal effects were observed at 10 nM. In silico modeling of binding of an arm of sv6D to human CLEC10A [68] predicted a RMSD of 1.343 Å and a binding energy of ΔG′ = −37 kJ/mol (Figure 5). A convincing feature of the mimicry was the binding of antibodies raised against the hexapeptide-KLH conjugate to GalNAc-PAA [67]. 

svL4 and sv6D bound with high specificity to CLEC10A in a lysate of human monocyte-derived DCs (Figure 5C). Biotinylated peptides, when attached to magnetic beads coated with streptavidin, retrieved the same protein from the lysate as an antibody against CLEC10A. We investigated with in silico modeling whether these peptides would also bind to the GalNAc-specific receptor MGL2 (CD301b), the mouse ortholog of CLEC10A. The sequences of these receptors are highly homologous (87% similar and 66.5% identical), with the highest identity in the sugar-binding domains. Because a crystal structure of MGL2 is not available, a homology model was generated with SWISS-MODEL [69] from the recently published crystal structure of human CLEC10A [68]. The predicted energy of binding of sv6D to the mouse receptor, ΔG′ = −35 kJ/mol, compared favorably with that for the human receptor, ΔG′ = −37 kJ/mol. These results suggest that the peptides bind with similar avidities to the mouse MGL2 and the human CLEC10A receptors, which validated the use of mouse models to investigate processes that involve this receptor. 

The human C-type GalNAc-specific receptor CLEC10A (CD301) is expressed most strongly on M2a macrophages and DCs [37,38]. The endocytic activity of CLEC10A has been well documented [70,71,72]. When a ligand binds, the cytoplasmic tyrosine-containing motif (YENF) is phosphorylated and the receptor is internalized [71]. CLEC10A is a calcium-dependent (C-type) lectin receptor, and internalization in response to a ligand causes an increase in the cytoplasmic concentration of Ca^2+^. The increase in Ca^2+^ initiates a signaling pathway that activates the phagocytic function of macrophages [38,73]. 

The role of CLEC10A as a modulator of the activities of DCs and macrophages has been extensively studied [38,48]. CLEC10A (MGL) is upregulated by inflammatory conditions in macrophages and brain microglia [74,75,76,77,78,79]. Valverde et al. [75] argued that the primary function of CLEC10A (MGL) is to provide a protective response to persistent inflammation and allow tumor cells to escape the immune system by the binding of tumor-expressed Tn antigens, i.e., a tolerogenic response. Zaal et al. [76] prepared dendrimers with 16 surface amino groups, to which Gal, GalNAc-Gal, or GaNAc-Gal-Glu were linked by reductive amination. Because this reaction involves the reducing end aldehyde group of the glycan, the sugar linked to the dendrimeric structure would be in the “open” conformation. Binding of nanomolar concentrations of MGL demonstrated high avidity of the GalNAc-dendrimer to CLEC10A. When monocyte-derived DCs were added to microtiter well coated with the dendrimers, production of IL-10 was stimulated but only in the presence of lipopolysaccharide (LPS) [76]. IL-10 is an immune suppressive and anti-inflammatory cytokine and thus an important factor in generating the tolerogenic state. 

An alternate view of CLEC10A function emerged from dose–response studies of treatment of murine models of ovarian cancer and glioblastoma with peptides (Figure 6 and Figure 7). The response elicited by a receptor is often determined by the degree of occupancy by a ligand. High occupancy, as provided by high ligand concentrations or the saturation binding of an antibody to the receptor, causes tolerance. This response results from activation of a signal transduction pathway that leads to phosphorylation of the transcription factor CREB to produce IL-10. In contrast, low occupancy activates a pathway that leads to NF-κB and production of IL-12 [48]. Studies of the interactions of ligands and receptors as a function of occupancy indicated that at low ligand/receptor ratios (5 to 20% receptor occupancy), receptor cross-linking is favored, and the ligand binds with high avidity because of favorable binding entropy. However, cross-linking at high ligand/receptor ratios (greater than 50% receptor occupancy) is less likely, and consequently, the initiation of signaling pathways is weaker [35]. 

Similar pharmacodynamic curves were obtained with svL4 and sv6D in very different disease conditions (Figure 6). The inflammatory environment of ascites in the peritoneal cavity of mice with ovarian cancer recruits monocytes, which differentiate into DCs and macrophages. These cells efficiently cross-present antigens to T cells, but only monocyte-derived DCs induce the differentiation of cytotoxic CD8^+^ T cells [80]. Experiments in which sv6D was injected subcutaneously every other day showed a significant extension of survival of mice with implanted ovarian cancer cells, essentially equal to that supported by the chemotherapeutic drug, paclitaxel [67]. Because only monocyte-derived DCs express a significant level of CLEC10A in the peritoneal environment [80], we interpreted these data as showing that activation of DCs by sv6D led to an enhanced antitumor state. In this system, a dose of 1 nmol/g body weight was only 10 to 20% as effective in extending survival as 0.1 nmol/g. 

A similar response to dose was observed when svL4 was injected subcutaneously into mice in which a glioma cell line had been implanted in the brain (Figure 7). In this study, circulating peripheral blood monocytes increased more than four-fold, and a two- to three-fold increase in the number of macrophages was found in the tumor and peritumoral areas in the brain with injections of svL4 [77,78]. The expression of MGL2, the target for svL4 on macrophages, is increased several-fold in inflamed tissue around the tumor [74,79]. Dusoswa et al. [79] showed that human glioblastoma tumor cells expressed an elevated number of surface Tn antigens, which was associated with increased levels of MGL expressed by macrophages in the tumoral area. These authors suggested that prognosis is poor because MGL is immunosuppressive. This effect may result from the higher number of Tn antigens, which leads to high receptor occupancy and subsequent tolerance, as shown in Figure 6. However, as shown in Figure 7, a low dose of 0.1 nmol/g body weight svL4, a Tn mimetic, strongly inhibited tumor growth. We interpret the data in Figure 7 to suggest that a high avidity, small-molecule ligand such as svL4 activates monocyte-derived cells that migrate into the tumoral area and provide a therapeutic response. An in vitro analysis of phagocytic activity of macrophages in a tumor explant showed that cells treated with 10 nM svL4 internalized bacterial cells, whereas untreated control cells were inactive.


### 3.4. E-Selectin and HNK-1

E-Selectin is a C-type lectin cell adhesion receptor that specifically recognizes sialo-fucosylated Lewis structures present on leukocytes and tumor cells. IELLQAR, a peptide that inhibited colonization of tumor cells, was discovered by screening a 7-mer phage display library [81]. Interestingly, no phage was selected when E-selectin was used in the screen because of low affinity (IC_50_ = ~750 μM), but peptides that bound to E-selectin were found by screening with a panel of anti-carbohydrate antibodies. The cyclic peptide CIELLQARC bound to E-selectin with a IC_50_ = ~100 μM, whereas an octavalent structure bound with an IC_50_ = 10 μM. An analysis of the modifications of the sequence led to two variant peptides, C-(IEELQAR) (K_D_ = 35 μM) and C-(IELFQAR) (K_D_ = 16 μM) [82].

Phage display libraries were used to search for a mimetic of the trisaccharide 3-*O*-sulfo-glucuronic acidβ(1-3)-Galβ(1-4)-GlcNAcβ, a glycan attached to the human natural killer cell-1 (HNK-1) [16]. The glycan is a ligand for the neural cell adhesion molecule (NCAM) and is crucial for regeneration of severed motor axons. The cyclic peptide C-(LSETTI) was found to have similar molecular properties as the glycan [17]. 

## 4. Peptides in Treatment of Neutrophilic Skin Diseases

Central to the homeostasis of the skin in the mouse are two related receptors, the Gal-specific CD301a (MGL1) and the GalNAc-specific CD301b (MGL2), the ortholog of human CLEC10A, that are expressed by M2a macrophages in the dermis. A serendipitous but important finding was an additional feature of our glycomimetic peptides that is essential in treatment of inflammatory skin diseases. Terminal differentiation of keratinocytes in the epidermis yields the stratum corneum, the first line of protection against environmental pathogens and allergens. Disruption of the stratum corneum, either by genetic deficiencies or physical/chemical insults, leads to inflammatory conditions of eczema caused by infiltration of neutrophils into the skin. LPS, proteins in house dust mites (HDM), and Staphylococcus enterotoxin B are commonly encountered allergens that cause eczema [83]. CD301a (MGL1) binds Gal residues on the major allergen in dust mites [84], presumably leading to phagocytosis and the destruction of the allergen by macrophages. Kanemaru et al. [85] found that the NC/Nga strain of mouse has a loss-of-function mutation in the gene encoding CD301a (MGL1, Clec10a) and is highly susceptible to LPS- and dust mite-induced infiltration of neutrophils, the primary cause of dermatitis. A high molecular weight glycoprotein that is rich in terminal T (Galβ(1-3)-GalNAcα1-*O*Ser/Thr) and Tn (GalNAcα1-*O*Ser/Thr) antigens was purified from HDM. This glycoprotein inhibited LPS-induced eczema in the tape-stripped skin of wild-type but not of *Clec10a^-/-^* mice. This finding indicated that CD301a^+^ macrophages are a major cell type in the defense against allergens and pathogens. Dupasquier et al. [86,87] found that half of the nucleated cells in the dermis of mouse skin were positive for antibodies against CD301a and CD301b, a characteristic of M2a macrophages. Gene regulatory network modeling of polarization indicated that M2a is the most frequent macrophage phenotype [88]. Efferocytosis (phagocytosis of apoptotic cells) of neutrophils by macrophages [89,90,91,92] is a major event in the resolution of inflammation and restoration of tissue homeostasis. 

The HDM allergen [85] induces an inflammatory response through toll-like receptor 4 (TLR4). LPS is a well-documented ligand for TLR4, mediated by CD14, which activates a pathway leading to the transcriptional factor NF-κB. We induced dermatitis on wild-type C57BL/6J mice with a combination of 1% SDS and 10 μg/cm^2^ LPS and tested whether the peptide mimetics of GalNAc, svL4 and sv6D, would ameliorate neutrophilic-driven eczema similar to that of the glycoprotein described by Kanemaru et al. [85]. Structural disruption of the epidermis was completely repaired within 14 days when 1 μM svL4 was applied topically along with LPS [93]. Restoration of the surface barrier also led to the elimination of neutrophils from the dermis. sv6D was more effective than svL4 when administered subcutaneously, particularly with severely damaged skin as characterized by neutrophilic dermatoses (unpublished results). Because CD301b^+^ (MGL2^+^) macrophages are essential for healing wounds in the skin [94,95], the higher avidity of sv6D with the receptor and its smaller molecular size suggest greater access to the dermis and thus promote this peptide as the superior drug for treating eczema and more serious dermatitis diseases. 

LPS-induced inflammation leads to a compensatory inhibition by IL-10 [96,97]. IL-10 production is also induced through TLR4 but by a pathway that activates the transcription factors CREB (cAMP response element binding protein) and ATF1 (cAMP-dependent transcription factor 1) [96]. Inhibition of transcription of LPS-stimulated genes is the primary mechanism of IL10-mediated suppression of inflammation in macrophages [98,99]. In mice deficient in the IL-10 receptor, an LPS challenge can be fatal [100]. Relevant to dermatitis, IL-10 is produced by macrophages, DCs, and neutrophils [96,99]. A seminal observation by Zaal et al. [76] was the strong stimulation of IL-10 production in monocyte-derived DCs by a multivalent GalNAc-dendrimer. We therefore concluded that a possible role of svL4 and sv6D, as mimetics of GalNAc, in rapid restoration of neutrophilic dermatitis was stimulation of IL-10 production by engagement of the receptor CD301b (MGL2) expressed by macrophages in addition to their phagocytic activity that was stimulated by an increase in cytosolic Ca^2+^.

svL4 contains glutamine residues that are an α-helical turn distant from each other and is a functional substrate of transglutaminase 2 (TGM2), also called tissue transglutaminase. Peptide sv6D, the C-terminal half of svL4, contains a glutamine residue at position 2 and is also a substrate for TGM2 [93]. Transglutaminases are critically important for functional development of the skin, and TGM2 and other isoforms are strongly induced by inflammatory conditions when the skin is damaged by wounds [101,102,103] or disease [104,105]. TGM2 catalyzes the formation of ε(γ-glutamyl)lysine iso-peptide bonds between glutamine and lysine residues, but the paucity of available glutamine residues at the skin surface seems to limit the ability of the enzyme to tighten the surface barrier and prevent transepithelial loss of water. Thus, healing occurs slowly. The topical application of the peptides provided abundant glutamine residues as substrates for cross-linking reactions. In this mouse model of eczema, the peptide rapidly restores the epidermis to normal morphology and promotes the elimination of the drivers of inflammation, neutrophils, from the dermis. 

TGM2 occurs within cells that express the enzyme but is also secreted by a non-classical mechanism [106]. In the cytosol, TGM2 exists in an inactive “closed” conformation in which the catalytic site with the reactive cysteine residue is covered by domains β-barrel1 and β-barrel2, which hinders access of the substrate. The closed conformation is stabilized by GTP or GDP bound to β-barrel1 (within the circle in Figure 8). In the extracellular environment, TGM2 undergoes a dramatic conformational change, from the closed form to a fully ‘open’ form in which Ca^2+^ displaces the guanine nucleotide and stabilizes the active enzyme [107]. The open form of the enzyme requires millimolar concentrations of Ca^2+^ for activity [108]. As shown in Figure 8, bound to the catalytic site is an inhibitory peptide, a derivative of a substrate, Ac-PQLPF-NH_2_, in which the glutamine is replaced with 6-diazo-5-oxo-**L**-norleucine [107], that stabilizes the open form. In our experiments, the enzyme freshly dissolved from a lyophilized protein was active in an assay with svL4, the tetravalent peptide with 12-mer active arms plus a 4-mer linker to the tri-lysine core, but not with sv6D, which has an active 6-mer sequence. The K_m_ with svL4 as substrate was approximately 7 μM. However, after the solution was stored for 3 days at 5 °C, sv6D and svL4 were equally active. Possibly, the longer arm was able to gain access to the active site cysteine but access by the shorter arm was restricted by the remainder of the tetravalent structure. These results suggested that the enzyme structure slowly gained the open conformation with which access to the active site was less hindered. In a condition such as eczema, which persists for extended periods of time, TGM2 will likely exist as newly synthesized (closed) and aged (open) forms. Thus, a treatment with svL4 or sv6D should essentially be equally effective. 

In addition to the essential process of repair of the surface barrier, activation of dermal macrophages may also contribute to healing of the injury. As described by Shook et al. [94], macrophages that express MGL2 (CD301b) are essential for wound healing. With healthy skin, it is generally considered that molecules larger than 500 Da cannot penetrate the stratum corneum [109]. Formulation protocols for larger drugs and biomolecules have been developed for transdermal delivery [110,111]. However, eczema and neutrophilic dermatoses are characterized by a loose surface barrier, disruption of the epidermis, or even complete loss of the epithelium. In these cases, svL4 or sv6D could diffuse into the dermal regions of the skin, promote a reduction in neutrophils and inflammation as well as subsequent healing by engaging dermal macrophages. The peptides therefore appear to be excellent tools for the repair of damaged skin by a two-pronged mechanism. 

## 5. Concluding Remarks

Peptides are highly flexible in sequence and conformation. Discovery of mimetics of sugars by screening phage display libraries relies on a selection of a peptide that achieves a structure that fits the conformation of a sugar-binding site on a protein, an induced fit as described by Changeux [31] that incorporates peptide–protein strategies elucidated by London et al. [112]. Although in silico modeling is generally a powerful ally in understanding binding of ligand to a receptor, limitations are also illustrated in this article and must be confirmed by direct binding assays. The high avidity of multivalent peptides, afforded by ligand density and favorable entropy factors, allows the engagement of receptors on immune cells beneath the canopy of the cell surface glycocalyx. The response of cells is “tunable” by manipulating the occupancy of the ligand, with maximal responses at the lowest doses. Peptides therefore provide excellent tools that can be developed for a variety of therapeutic needs. These are lessons that will be valuable in ongoing research. 

## Figures and Tables

**Figure 1 pharmaceutics-15-00688-f001:**
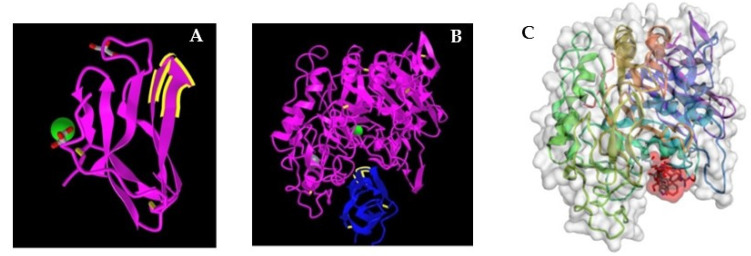
Structure of the complex of α-amylase with tendamistat. (**A**) The crystal structure of tendamistat (PDB accession no. 1OK0). The loop structure that binds to the glycan binding site of α-amylase is highlighted. (**B**) The crystal structure of the complex with tendamistat bound in the glycan binding site of porcine α-amylase (accession no. 1BVN). (**C**) Model of the 6-mer loop structure of tendamistat (shaded in red) bound to porcine α-amylase (accession no. 1PIF) generated in silico with CABS-Dock (RMSD = 3.561 Å). The predicted binding energy, ΔG′ = −56 to −60 kJ/mol, is nearly the value calculated from K_eq_.

**Figure 2 pharmaceutics-15-00688-f002:**
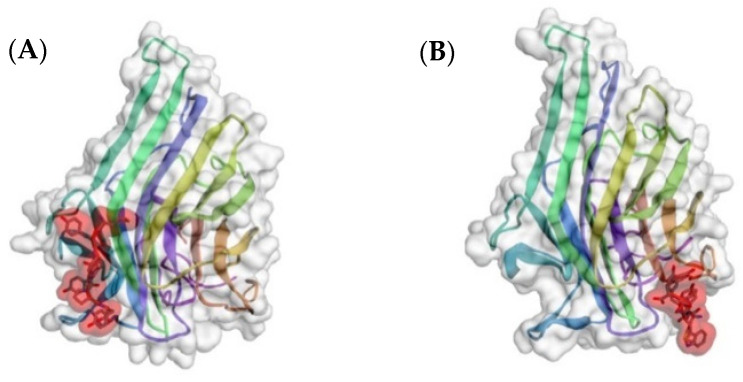
Binding of the peptide MYWYPY to ConA (accession no. 5YGM) as predicted by CABS-Dock. (**A**) A model of the peptide bound to the left side of the Man binding site (Model 2: RMSD = 5.621 Å; ΔG′ = −43 kJ/mol). (**B**) A model of the peptide bound to the right side of the Man binding site (Model 4: RMSD = 5.997 Å; ΔG′ = −43 kJ/mol). The peptide is shaded in red.

**Figure 3 pharmaceutics-15-00688-f003:**
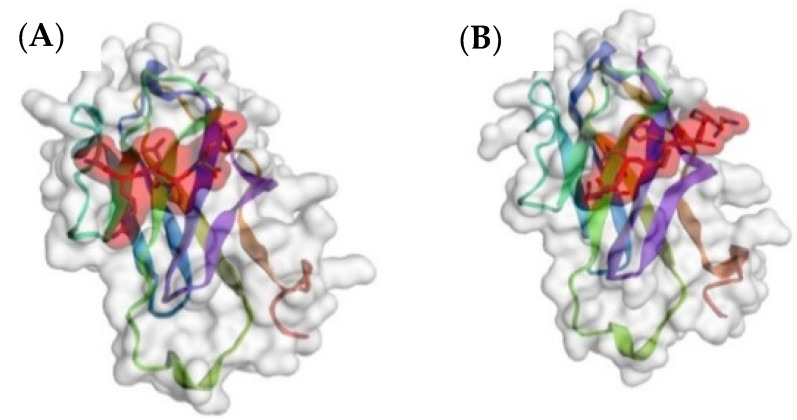
In silico comparison of predicted binding of peptides (**A**) ARLPR (RMSD = 2.331 Å, ΔG′ = −46.4 kJ/mol) and (**B**) NPSHPLSG (RMSD = 2.848, ΔG′ = −45.1 kJ/mol) to Siglec-7 (accession no. 1O7V). Modeling was performed with CABS-Dock. Peptides are shaded in red.

**Figure 4 pharmaceutics-15-00688-f004:**
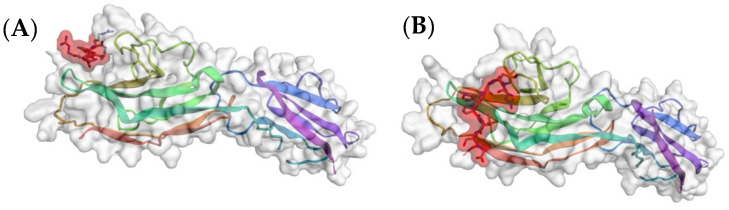
In silico comparison of binding of (**A**) ARLPR and (**B**) NPSHPLSG to Siglec-5 (accession no. 2ZG1) as predicted by CABS-Dock. The figure shows model 6 for ARLPR (RMSD = 14.47 Å, ΔG′ = −46.4 kJ/mol) and model 1 for NPSHPLSG (RMSD = 4.518 Å, ΔG′ = −44.3 kJ/mol). Peptides are shaded in red.

**Figure 5 pharmaceutics-15-00688-f005:**
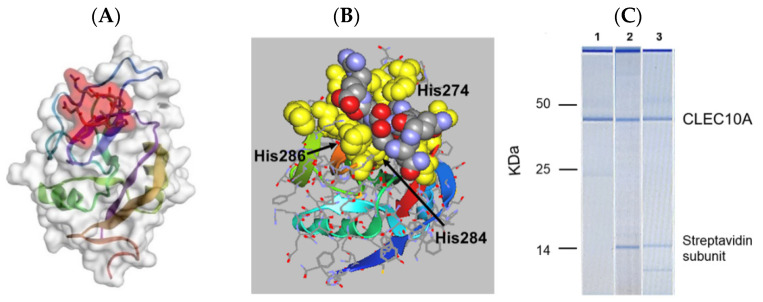
(**A**) In silico docking of an arm of sv6D (NQHTPR) to human CLEC10A (accession no. 6PY1, RMSD = 1.343 Å, ΔG′ = −37 kJ/mol) with CABS-Dock. The peptide is enclosed in red shading. (**B**) The structure in (**A**) was downloaded into ArgusLab. The position of sv6D in the binding pocket is shown after additional molecular dynamics. The peptide is colored (carbon, grey; nitrogen, blue; oxygen, red) while the binding site is yellow. The positions of His^274^, His^284^, and His^286^ of the binding site are indicated. (**C**) A lysate of human monocyte-derived DCs was incubated with (1) mouse anti-human CLEC10A, which was recovered with magnetic beads coated with protein A; (2) biotinylated sv6D; or (3) biotinylated svL4, which were recovered with magnetic beads coated with streptavidin. Proteins were eluted from the beads and subjected to electrophoresis. Molecular mass markers are indicated for IgG heavy chain (50 kDa), IgG light chain (25 kDa), and a streptavidin C1 subunit (13.6 kDa). The top band is an instrument marker.

**Figure 6 pharmaceutics-15-00688-f006:**
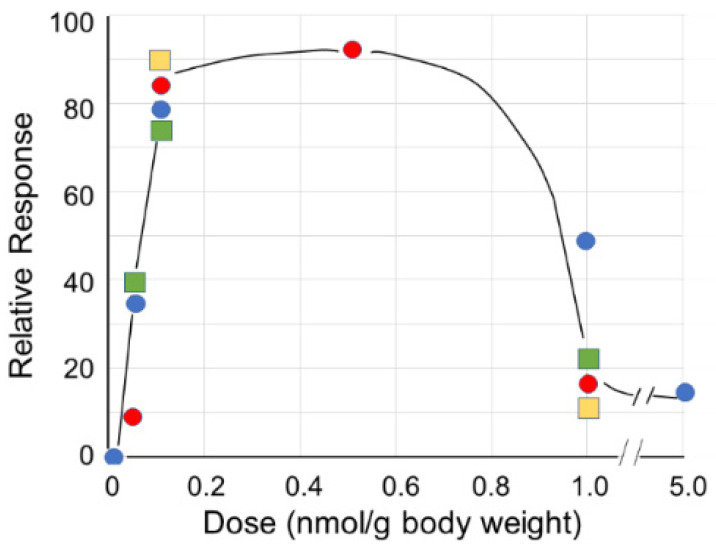
Effect of dose on response in C57BL/6 mice. The peptides were injected subcutaneously, and measured endpoints included proliferation of progenitor peritoneal cells in healthy mice (red circles), inhibition of growth of glioma tumors with cells implanted into the brain (blue circles), and survival of mice with implanted ID8 ovarian cells treated with svL4 (green squares) or sv6D (yellow squares). The response curve reflects the extent of receptor occupancy, with greater than 50% leading to inhibition or tolerance.

**Figure 7 pharmaceutics-15-00688-f007:**
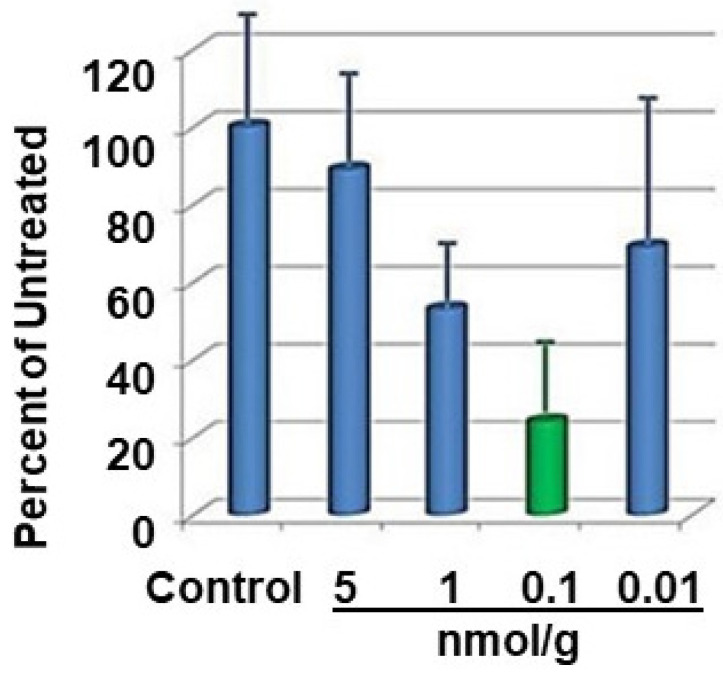
Growth of glioma tumor after implantation of GL261 glioma cell line into brain of C57BL/6 mice. svL4 was injected subcutaneously on alternate days, starting at day 7 after implantation, at the doses indicated. Tumor size was measured by NMR [77].

**Figure 8 pharmaceutics-15-00688-f008:**
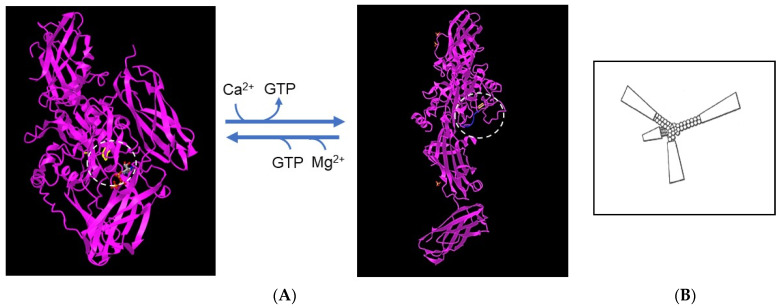
Structures of transglutaminase 2. (**A**) The inactive, closed conformation (accession no. 3LY6) and (**B**) the fully open, active conformation (accession no. 2Q3Z). The catalytic site of the enzyme is circled and the position of the cysteine residue that forms the thioester linkage to the substrate in the first step in the reaction is highlighted. The closed conformation is stabilized by Mg^2+^ and a GTP molecule, shown in gray, that binds to β-barrel1 and covers the catalytic site. Calcium displaces Mg^2+^ and GTP and generates the open conformation, which is stabilized by an inactive derivative of the peptide substrate. As shown within the box at the right, the remainder of the tetravalent structure of svL4 and sv6D may restrict entry of an arm into the catalytic site of the closed or partially open conformation.

## Data Availability

Not applicable.

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
