# Peer review of "Glycomimetic Peptides as Therapeutic Tools"

_pharmaceutics, 2023, doi:10.3390/pharmaceutics15020688_

Round 1

Author Response

We appreciate the constructive comments by the reviewer.  The manuscript has been extensively revised, with additional information regarding phage display and a rearrangement of sections of text to accommodate the reviewer's recommendation and to improve the flow of information.  

The typos in the text and errors in the reference section have been corrected.  

Reviewer 2 Report

The authors present an interesting overview of peptides that can mimic sugars in therapeutic applications. The topic is certainly an interesting one, and the authors give a comprehensive overview of the field. 

Overall, I have a few misgivings, which I believe should be addressed before publication in Pharmaceutics.

1. The scope of the review is unclear. By this, I mean that the abstract, introduction and conclusion do not define the requirements for a paper to be included in the review (i.e. normally a review would specify if it all papers, recent advances, a particular aspect, etc.). It should be noted a quick internet search shows papers in this area that are not included in the review. Therefore, I believe that the authors should include within the introduction a statement on how the discussed papers were selected.

2. Section 9 (concluding remarks) would benefit from a paragraph describing the authors opinion on the future directions for this field, and the future perspective.

3. Sections 2 and 3, while important for introducing the area, do not contain any mention of peptides as glycomimetics. I would therefore suggest that these two sections are incorporated into the introduction section as subsections.

4. The reader would benefit from some of the molecular structures of ('small') compounds being drawn in full. For example, the structure shown in line 56 requires the reader to decipher the positions of connectivity from the long chemical names in lines 53/54, and to look up the structure of sialic acid. Therefore, I would suggest that the chemical structure should be shown.

5. There are some typographical/formatting issues.

- For example, there seems to be large gaps after full-stops.

- Minus signs and number ranges should be en dash instead of hyphen signs.

- There is some unnecessary capitalisation, such as Galectins in line 68.

- There is an equals sign (=) in line 118 that should be a hyphen.

- The equals sign (=) in line 189 is subscripted.

- The formatting of the reference section needs to be addressed.

Author Response

We appreciate the constructive comments by the reviewer.  The manuscript has been extensively revised, with additional information regarding phage display and a rearrangement of sections of text to accommodate the reviewer's recommendation and to improve the flow of information.  Additional papers have been included and the overall intent of describing the activity of glycomimetic peptides has been expanded.

Reviewer 3 Report

This manuscript is a comprehensive review article about glycomimetic peptides. The contents are interesting and contribute to the field of peptide-based therapeutics. I have no major comments except a minor comment. It is recommended to provide a Table that summarizes the main peptides (sequence, references, key features, etc.) described in each section.    

Author Response

We appreciate the reviewer's comments.   Many of the peptide sequences in each section describe the progression in the search for the optimal sequence.  Only a few peptides reach the status of a "druggable" product, and thus we did not consider the number sufficient for collating the information into a table.